# Feasibility Study on Geothermal Dolomite Reservoir Reinjection with Surface Water in Tianjin, China

Donglin Liu [1,2], Yun Cai [3], Zhaolong Feng [1,2], Qiuxia Zhang [1,2,*], Lisha Hu [1,2] and Shengtao Li [1,2]

1  Center for Hydrogeology and Environmental Geology Survey, China Geology Survey, Tianjin 300304, China; liudonglin@mail.cgs.gov.cn (D.L.); fengzhaolong@mail.cgs.gov.cn (Z.F.); hulisha@mail.cgs.gov.cn (L.H.); list07@mails.jlu.edu.cn (S.L.)
2  Tianjin Geothermal Exploration and Development Design Institute, Tianjin 300304, China
3  Tianjin Geothermal Resources Exploration and Development Engineering Research Center, Tianjin 300304, China; caiyunjiayou@126.com
*  Correspondence: edu652524@163.com

**Abstract:** Reinjection is thought to be the most effective way to maintain reservoir pressure and production capacity for hydrothermal resources. The use of external water injection to replenish deep geothermal reservoirs is a new approach in China to addressing the problems of declining groundwater levels and energy depletion caused by the excessive and uneven exploitation of geothermal resources. However, the key challenge and focus of the feasibility assessment of this method lies in the chemical compatibility of the external water with the native geothermal reservoir water and surrounding rocks. In this paper, we discuss the geochemical response of a dolomite reservoir to lake water injection based on experiments on water–rock interaction in the Wumishan formation in the Dongli Lake area of Tianjin. The results show that after reactions with dolomite, the TDS of the reacted water decreases, indicating the occurrence of precipitation. According to the calculation results obtained using the PHQREEC program, the precipitation amount is found to be quite limited. Geochemical analysis indicates that at the initial stage of the reactions, plagioclase dissolves and releases alkaline metals like Ca-, Na-, SiO2- and Al-bearing compositions, leading to the oversaturation and precipitation of dolomite and calcite. As the reaction progresses, a portion of the dolomite dissolves, while the calcite continues to precipitate at a later stage. Illite precipitates and its effects on reservoir structure depend on its shape. Based on the experimental data, it can be concluded that the dolomite reservoir will be slightly affected by the reinjection of lake water; however, it is still a good method for the sustainable development of geothermal resources.

**Keywords:** dolomite geothermal reservoir; lake water reinjection; water–rock interaction; PHQREEC



## 1. Introduction

As a kind of renewable green energy, the efficient and reasonable development and utilization of geothermal resources is of great significance for energy conservation, emission reduction and haze mitigation in northern China [1–4]. Medium–low-temperature geothermal resources are abundant and widely distributed in China, and their direct utilization ranks first in the world [5]. For a long time, however, geothermal resource development in most areas has been plagued by problems such as single wells or unbalanced production and injection, resulting in a rapid decline in groundwater levels. According to statistics, the water level in the Neogene, Ordovician and Jixian systems can be decreased by as much as 10 m every year [6,7]. At present, recharge has become an effective mode to achieve the sustainable development and utilization of geothermal resources [8,9], and it has been widely used in more than ten countries, such as Italy [10], the United States [11], New Zealand [12], Iceland [13], France [14], Japan [15] and Denmark [16]. Geothermal reinjection in China began in the 1970s, and researchers have carried our research in the Beijing Xiaotangshan geothermal field [17], Tibet Yangbajing geothermal field [18], Shandong

northwest depression area [19,20], Hebei Handan city [21], etc. Research on the recharge of bedrock fissure geothermal reservoirs in Tianjin began in the 1980s. In 1997, researchers in Tianjin carried out a special scientific recharging test at the Wanglanzhuang geothermal field using non-thermal periods. At the beginning, the study of recharge in Tianjin was mainly combined with a productive recharge operation during the heating period. Since 2005, geothermal reinjection work in Tianjin has entered the phase of regular production and operation. The reinjection system has been evolving and the overall reinjection rate has been increasing year by year, and related reinjection technology development has become more and more scientific and reasonable. In geothermal management, planning work has also begun to use advanced information management means and methods, introducing a series of regulations and standards, and geothermal recharge work has gradually entered the stage of rapid development [22–24].

In the past, most geothermal reinjection studies used the cooling tail water as the reinjection water source [25–27]. The reinjection of external water into geothermal reservoirs has been proposed as a new geothermal development strategy. The aim is to alleviate or even suppress the decline in geothermal water levels by artificially replenishing geothermal water resources, thereby increasing the amount of geothermal resources that can be recovered and protecting geothermal clean energy sources. This approach has been applied to oil extraction. This has a positive effect on avoiding environmental problems such as land subsidence and water level decline [28,29]. In the Dongli Lake area, recharge experiments have also been carried out using external water sources, and some experience has been obtained [30–32]. However, in the practice of geothermal resource development and utilization engineering, there are still only a few reports on the recharge of geothermal reservoirs by external water recharge [33–36]. The main concern is that the introduction of exogenous water may cause complex geochemical reactions with the water in the original geothermal reservoir and the surrounding rock, such as water pollution, reservoir damage, wellbore instability, permeability decline, etc. [37–39]. These reactions not only affect the extraction efficiency and stability of geothermal energy, but also may have a potential impact on the geological environment [40–50]. In this paper, we conduct water–rock interaction experiments at the Wumishan Reservoir in Jixian District, Dongli City, Tianjin, and discuss the recharge feasibility of untreated lake water, the response of the reservoir and its impact on the reservoir.

## 2. Materials and Methods

### 2.1. Overview of the Study Area

The research area is situated in the north–central segment of the Cangxian Uplift, which is a part of the North China Platform. Primarily located in the central region of Tianjin, the area is divided by the Ninghe–Baodi Fault into the northern mountainous terrain and the southern plain region, each exhibiting distinct geological structural characteristics. The northern mountainous terrain belongs to the Jibao Uplift Fold Zone, a tertiary tectonic unit (Grade III) within the Yanshan Platform Fold Zone, which is a secondary tectonic unit. Here, formations of the Paleozoic and pre-Paleozoic eras are prominently developed. The structural trends are mainly oriented in an east–west (EW) direction, with EW-trending faults being the most dominant. Additionally, faults trending in a northwest (NW), northeast (NE) and north–northeast (NNE) direction are also observed. These fault structures play a pivotal role in controlling the distribution and exposure patterns of geological strata within the mountainous terrain. Conversely, the southern plain region pertains to the North China Fault Depression Zone, a secondary tectonic unit. This region is characterized by Mesozoic and Cenozoic fault-bounded and depressed basins. Within this zone, the tertiary tectonic units (Grade III) include the Cangxian Uplift, the Jizhong Sub-basin and the Huanghua Sub-basin. The Cangxian Uplift, together with the Jizhong Sub-basin and Huanghua Sub-basin, as well as numerous Grade IV tectonic units (such as uplifts and depressions) distributed among them, extend in an NNE direction, aligning in an echelon pattern. The larger faults also trend in an NNE direction, contributing to the

overall structural layout of the region (Figure 1). The pre-Cenozoic basement strata in the research area generally slope gently towards the northwest, consisting mainly of the Middle and Upper Proterozoic and Paleozoic formations. Overlying these strata is the Cenozoic formation, which is approximately 1400 to 1900 m thick and lacks the Paleogene series. Based on the average temperature gradients observed in each geothermal well within the study area, it is situated in the high-heat-flow region of the Cangxian Uplift. The average geothermal gradient of the cap rock is recorded to be 3.0 to 5.4 °C per 100 m. Notably, the highest geothermal gradient of the cap rock is found in proximity to the Cangdong Fault, which serves as a conduit for water and heat from deeper geological layers. As the distance from the Cangdong Fault increases, the geothermal gradient decreases gradually.

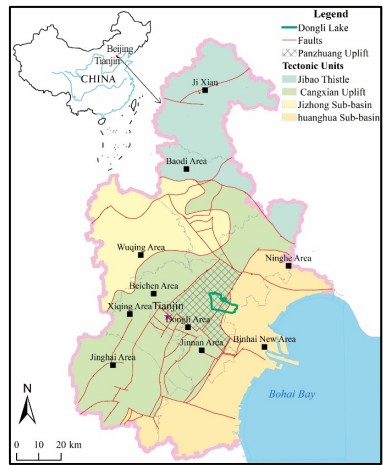 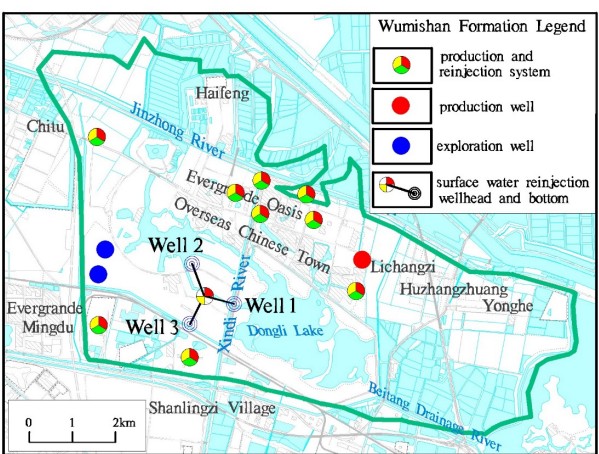

**Figure 1.** Location of study area.

Researchers in Tianjin have completed the exploration of eight geothermal fields to date. Multiple geothermal reservoirs are developed in this area, and the Wumishan formation geothermal reservoir serves as the primary exploitation layer. Development primarily takes the form of "one production and one injection" paired wells. The thermal power of paired wells exploited for a single heating season ranges between 6 and 10 MW. These are widely distributed in the southern plain area of Tianjin, and the distribution area shallower than 4000 m is mainly located in the Cangxian uplift area east of the Tianjin fault, covering an area of 3562 square kilometers. The water inflow rate of single wells is 100–120 m$^3$/h, with a stable flow temperature at the wellhead of 80–100 °C. The average reinjection capacity of reinjection wells is 100–150 m$^3$/h (27.7–41.6 L/s/MPa). The main fluid chemical types are Cl·HCO$_3$·SO$_4$-Na, Cl·SO$_4$·HCO$_3$-Na and Cl·SO$_4$-Na. The geothermal reservoir lithology of the Wumishan formation mainly includes karst breccia, grain dolomite, sandy dolomite, algal dolomite and siliceous dolomite. The geothermal reservoir type is mainly of the fracture type, followed by the dissolution pore type, and most of the fractures and dissolution pores are semi-filled. At present, there are 18 geothermal wells, and the water quality type includes HCO$_3$, Cl, SO$_4{}^{2-}$ Na and HCO$_3$∎Cl-Na, with a pH value of 7.1~8.3 and salinity of 1600~2200 mg/L [29]. Geothermal water is mainly used for the heating, bathing, planting and aquaculture industries. According to statistics, 13 production wells and 11 recharge wells had been drilled in the study area by 2014, and the total amount of geothermal resources extracted was 2.708 × 10$^6$ m$^3$/a, the total recharge volume is about 1.947 × 10$^6$ m$^3$/a, and the geothermal heating area reached 1.92 × 10$^6$ m$^2$. The Dongli Lake region produces roughly 180 MW of thermal energy every year, covering a heating area of approximately 4.5 million square meters.

The Dongli Lake in the area formerly known as the Xindihe Reservoir was artificially excavated and constructed in the 1970s. With a water area of 80 square kilometers and a total reservoir capacity of 2200 × 10$^4$ m$^3$, it boasts abundant water during the wet season

and good water quality, providing natural advantages for the construction of a centralized surface water recharge experimental base.

*2.2. Experimental Design*

2.2.1. Sample Description

The rock sample used in this experiment was dolomite debris from the Wumishan formation, Jixian System, well No. 1 of the surface water recharge project (Figure 1), with a sampling depth of 2000 m. The scanning electron microscope (SEM) results show that in some areas, there are aggregates or ooids, which appear pale yellow with low relief under plane-polarized light. Under crossed polarizers, they are composed of extremely fine particles (0.005 mm), exhibiting weak optical properties, and are identified as clay minerals. These clay minerals fill the spaces within the aggregates and can also be observed to fill the spaces between carbonate grains (Figure 2). The instrument model for whole-rock quantitative analysis was Panalytical X'Pert-PRO, and the analysis Conditions were a voltage of 40 kV, a current of 40 mA, an X-ray Target of Cu and a measurement angle range of 5°–70°. The whole-rock XRD testing results show that the average mineral composition is 78.25% dolomite, 20.90% quartz, 0.70% Illite, and 0.15% plagioclase. The principal element analysis shows that its $SiO_2$ content is 34.03%, MgO is 13.91%, CaO is 19.58% and $TFe_2O_3$ ($TFe_2O_3$ represents a broader concept, encompassing all iron elements that may exist in the form of iron oxides, regardless of their specific chemical forms) is 1.94%, with small amounts of $Al_2O_3$ (0.59%) and $K_2O$ (0.1%), and the loss on ignition (LOI) reaches 30.18%. Water samples were collected from Dongli Lake 10 cm below the surface with 20 mL, 100 mL and 1000 mL HDPE bottles, where the coordinates were E 39°10′24″, N 117°27′36″. The bottles were cleaned three times with the water samples before sampling. The regular components and trace elements of the sample water were tested and analyzed in the laboratory. The pH value of the lake water was 8.3, the TDS wa 1618 mg/L and the hydrochemical type was Cl·$HCO_3$-Na. The calculation results show that the carbonate minerals (including dolomite, calcite and aragonite) in this water are oversaturated.

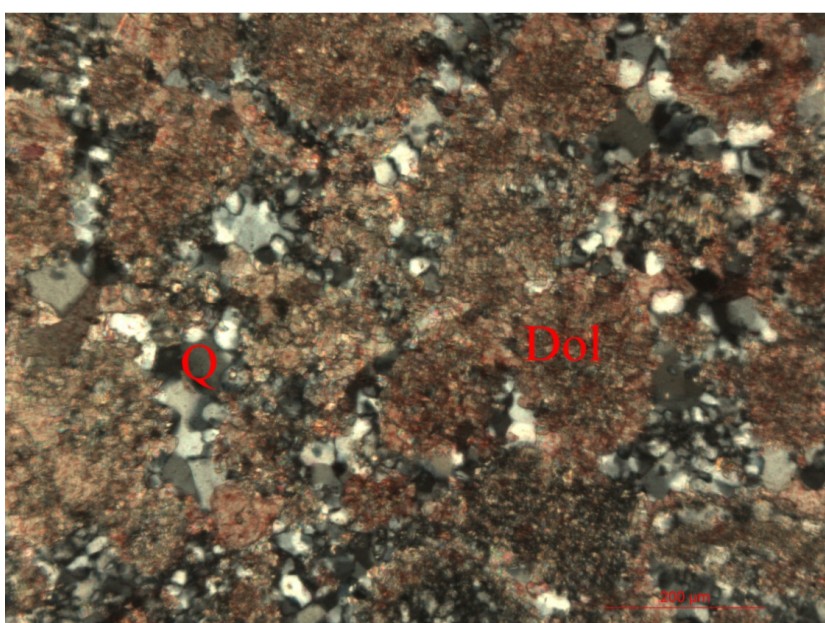

**Figure 2.** A scanning electron microscope image of the sample. The "Q" represents quartz and "Dol" represents dolomite.

2.2.2. Pre-Treatment of Water Samples Before Experiments

First, the water from Dongli Lake is pumped up and, during transport, a bactericide is added to the pipes to kill microorganisms, bacteria and other pollutants in the water. The

water then goes into a sloping-tube sediment tank for mud–water separation. This process causes sediment, suspended solids, colloids and microorganisms in the water to form large flocculent precipitates (aluminum floc), which are separated from the water. The water then enters an inclined-tube sedimentation tank for mud–water separation. The sludge is periodically discharged into a sludge tank, concentrated in a sludge-thickening tank, and filtered under pressure by a filter press. The clarified water is then discharged, while the sludge is transported externally for landfill disposal. The effluent from the sediment tank enters an intermediate tank and is pumped by a water supply pump into a sand filter tank to remove suspended solids and other impurities. The water is then passed through a deoxygenation tank to remove most of the dissolved oxygen before entering a precision filter to further remove suspended solids, colloids and other impurities. A portion of the pretreated water overflows into a recharge tank, while the remaining water enters a booster pump. The water from the booster pump is pressurized and then sent to nanofiltration (NF) equipment for membrane filtration. The filtered water enters the recharge tank and mixes with the overflow pretreated water, resulting in a water quality that basically meets the recharge water standards. To address the potential issues of pH and dissolved oxygen not meeting the standards, a small amount of acid (or alkali) solution and a deaerating agent can be added before recharge, based on the actual conditions, to ensure that the recharge water meets the required indicators. The treated recharge water is then transported by a recharge pump to the recharge well and ultimately injected back into the geothermal reservoir. A flow diagram is shown in Figure 3.

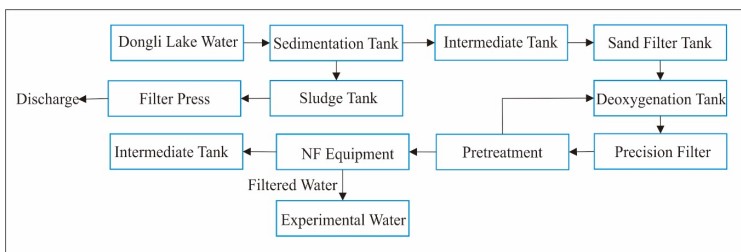

**Figure 3.** Process diagram for pre-treatment of experimental water.

### 2.2.3. Experimental Setup and Conditions

In order to simulate the water–rock interaction under reservoir temperature and pressure conditions as much as possible, a modified Parr model of a high-temperature and high-pressure reactor device (Parr 4575A) from the Parr Company in Moline, IL, USA was used in this experiment, as shown in Figure 4. This device is mainly composed of a reaction chamber (500 mL), a control system (pressure, temperature and stirring speed controllers) and a booster pump.

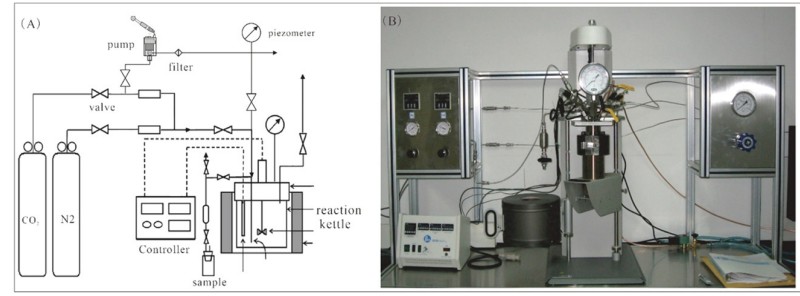

**Figure 4.** The experimental principles (**A**) and water–rock–$CO_2$ reaction equipment (**B**).

In order to simulate the chemical reaction process between lake water and rocks over a long period of time within a short experimental duration, we increased the specific surface area for water–rock reaction by using rock debris. Additionally, we employed stirring

bars within the reactor to accelerate the reaction rate between water and rocks, thereby achieving a full reaction in the laboratory within a short period of time.

In this experiment, the mass ratio of water and rocks was set at 6, and the masses of the water and debris samples were 300 g and 50 g, respectively. In order to discuss the effect of lake water recharge on the reservoir at different time scales, the reaction time series were set at 5 days, 10 days and 20 days. The temperature and pressure were set at 90 °C and 20 MPa according to the reservoir conditions corresponding to the sampling depth of the debris, and the stirring speed was 200 RPS. The temperature and pressure were monitored during the experiment.

### 2.3. Collection and Test Analysis of Reaction Samples

The water samples (RL1) were loaded into 20 mL and 100 mL sample bottles before the reaction, and the pH, TDS, complete hydrochemical analysis and hydrogen and oxygen isotopes were analyzed. The debris samples were ground to 200 mesh and loaded into sample bags for the analysis of mineral components and principal elements. When the reaction was carried out for 5 days, 10 days and 20 days, the device was cooled to room temperature and the samples sampled separately (RL2: reaction after 5 days; RL3: 10 days; RL4: 20 days), the water sample was filtered into 20 mL and 100 mL sample bottles for water chemistry and isotope analysis, and the pH, TDS and $HCO_3^-/CO_3^{2-}$ were measured immediately (Table 1). The debris sample in the chamber was collected on a clean glass dish and dried at 40 °C for 24 h in an oven. Then, the debris was ground to 200 mesh for analysis.

**Table 1.** Water chemistry of lake water for reinjection (unit: mg·L$^{-1}$) (RL1: original surface water; RL2: reaction after 5 days; RL3: reaction after 10 days; RL4: reaction after 20 days).

| Sample ID | pH | F$^-$ | Cl$^-$ | SO$_4$$^{2-}$ | HCO$_3$$^-$ | Na$^+$ | K$^+$ | Mg$^{2+}$ | Ca$^{2+}$ | SiO$_2$ | TDS |
|---|---|---|---|---|---|---|---|---|---|---|---|
| RL1 | 8.2 | 4.0 | 498.1 | 271.2 | 438.7 | 484.4 | 35.0 | 47.5 | 41.3 | 5.6 | 1618.3 |
| RL2 | 7.4 | 4.6 | 503.3 | 287.2 | 234.3 | 484.0 | 35.3 | 12.8 | 28.2 | 31.0 | 1503.6 |
| RL3 | 8.1 | 4.1 | 502.9 | 288.9 | 210.3 | 481.4 | 36.8 | 10.7 | 29.6 | 33.4 | 1493.1 |
| RL4 | 8.2 | 2.9 | 479.8 | 286.5 | 276.4 | 464.0 | 34.0 | 18.0 | 41.0 | 11.3 | 1475.7 |

### 3. Results and Discussion

#### 3.1. Geochemical Response of Geothermal Reservoir Fluids

The experimental results show that the variety of pH, TDS and hydrochemical components is regular. The content of Na$^+$, Cl$^-$ and K$^+$ was slightly decreased, while the content of SO$_4$$^{2-}$ was increased, but the increment was small. The content of Mg$^{2+}$, Ca$^{2+}$ and HCO$_3$$^-$ decreased at first (t = 5 days, 10 days) and then increased (t = 20 days), and the content of Ca$^{2+}$ was basically returned to the initial value (Figure 5a,b). The content of SiO$_2$ was increased at first and then decreased, indicating that chemical reactions caused the release of Si from the mineral into the water. The pH value increased at first and then decreased a little (Figure 5c), which was consistent with Mg$^{2+}$, Ca$^{2+}$ and HCO$_3$$^-$. The variation trend of HCO content was consistent (Figure 5a). It indicated that the change in pH was caused by chemical reactions. Among the three batches of reaction (t = 5, 10 and 20 days), TDS decreased continuously (Figure 5c), indicating that part of the material in the geothermal water was precipitated out in the form of precipitation. TDS reduced by 142.6 mg/L in total, which was equivalent to a 42.8 mg debris increase, accounting for 0.09% of the debris involved in the reaction. The influence of this amount on the reservoir structure was basically negligible.

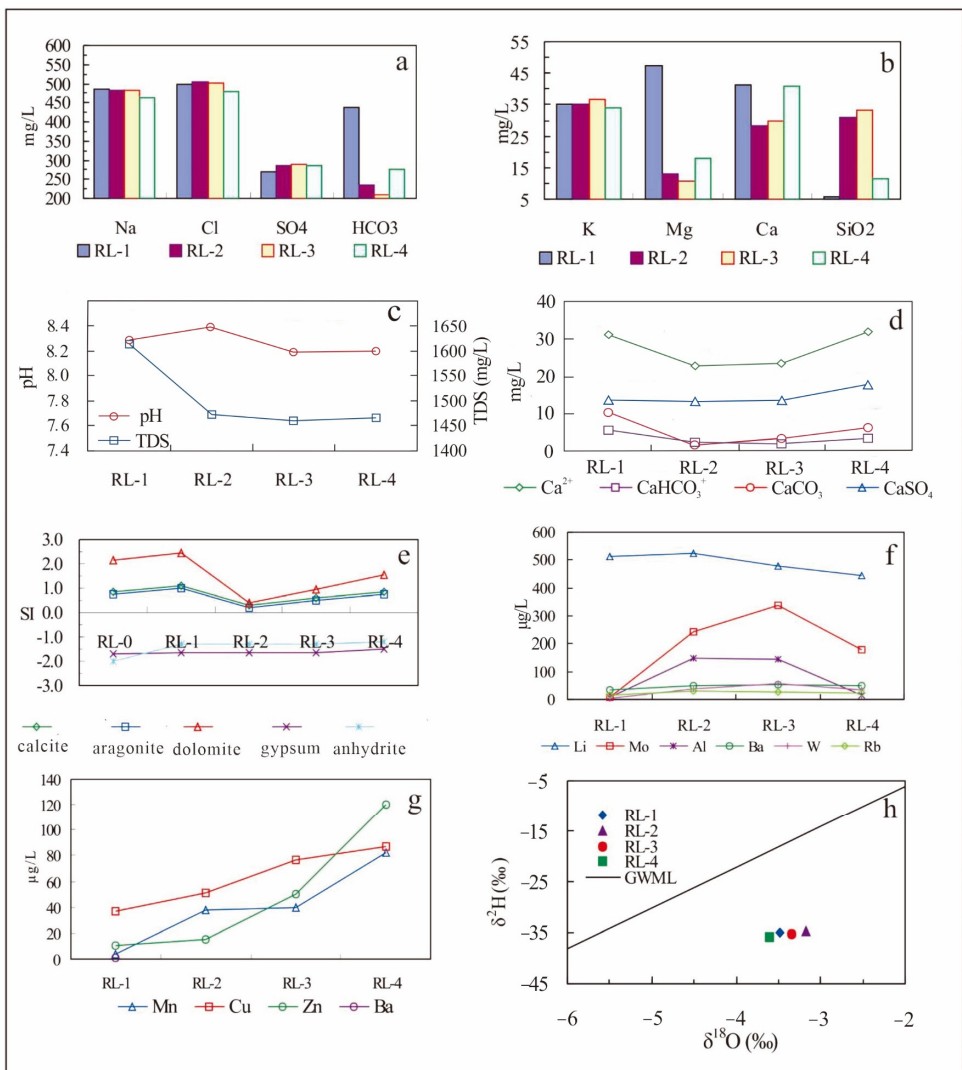

**Figure 5.** Water chemistry and isotope variations before and after reactions with different times. Figures (**a**,**b**) depict the variations in the primary ion content of the samples, while figure (**c**) outlines the trends in pH and TDS. Figure (**d**) highlights the shifts in the primary forms of Ca element content in the samples before and after the reaction. Additionally, Figure (**e**) shows the trends in saturation indices of the main minerals. Figures (**f**,**g**) present the changes in the concentrations of various trace elements, and Figure (**h**) illustrates the trends in hydrogen and oxygen isotope ratios.

The PHREEQC program was used to calculate the ion morphology and saturation index of geothermal water samples before and after the reaction. The results showed that the main forms of Ca elements included $Ca^{2+}$, $CaCO_3$, $CaSO_4$ and $CaHCO_3^+$. When the reaction times were 5 days, 10 days and 20 days (Figure 5d), the content of $CaSO_4$ was always increasing. $Ca^{2+}$, $CaCO_3$ and $CaHCO_3^+$ decreased first and then increased, and gradually recovered to the initial state.

Under the condition of reservoir temperature (90 °C), gypsum ($CaSO_4 \cdot 2H2O$) and anhydrite ($CaSO_4$) were always undersaturated (SI < 0), but anhydrite showed a tendency to precipitate. Carbonate minerals were always supersaturated, in which dolomite ($MgCa(CO_3)_2$) showed a dissolving trend and aragonite ($CaCO_3$) showed a tendency to dissolve at first and then precipitate (Figure 5e).

Different trace elements exhibit different reaction characteristics, as shown in Figure 5e,g. The elements Li, Mo, Al, Ba, W and Rb all display a trend of increasing first and then decreasing, which is consistent with the trend of $SiO_2$ but opposite to that of $Mg^{2+}$, $Ca^{2+}$ and $HCO_3^-$. Generally, lithium is distinctly incorporated within clays (such as mica and illite)

and feldspar. Subsequently, the lithium concentration decreases inversely with the increase in precipitation of K-feldspar and illite (as shown in Figure 6b). Similarly, the absence of barium in the fluid (depicted in Figure 5g) can be correlated with the precipitation of K-feldspar minerals, which typically trap barium. Among them, the content of Mo changes significantly, increasing from an initial 8.8 μg/L to 348 μg/L and then decreasing to 179 μg/L. The specific mechanism behind this change remains to be discussed. In contrast, the contents of Mn, Cu and Zn continue to increase, indicating that the chemical process promotes the release of these heavy metal elements from the rock debris samples.

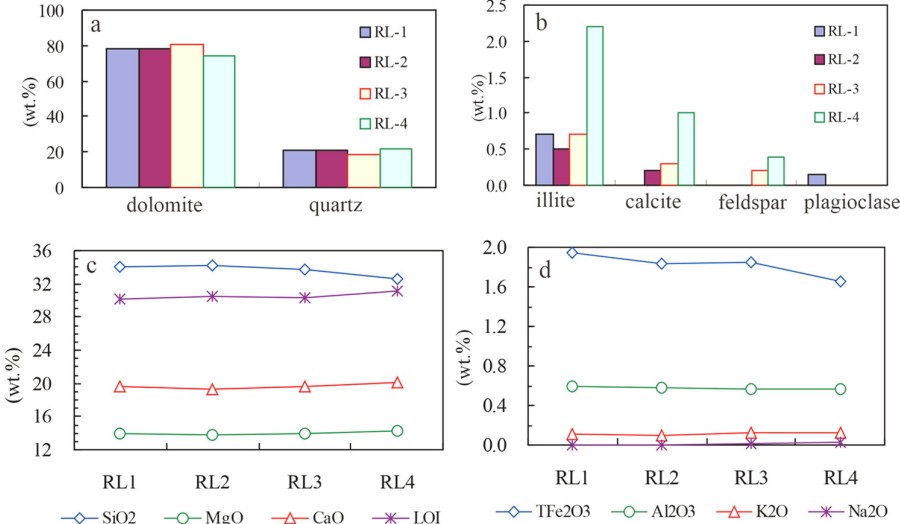

**Figure 6.** Changes in mineral components and major elements of carbonate before and after reactions. Figures (**a**,**b**) illustrate the changes in the content of major minerals before and after the reaction, while figures (**c**,**d**) show the changes in the content of major elements before and after the reaction.

The results of hydrogen and oxygen isotopes (Figure 5h) showed that the oxygen isotopes of geothermal water were enriched first and then depleted as the reaction proceeded. This is because the dolomite with relatively enriched oxygen isotopes dissolved and exchanged with the oxygen isotopes of geothermal water, making the oxygen isotopes of geothermal water more enriched than before. With the progress of the reaction, carbonate minerals began to precipitate, resulting in the dilution of oxygen isotopes in the water, which were basically restored to their initial state, showing the same trends for pH value and the water chemical components $Mg^{2+}$, $Ca^{2+}$ and $HCO_3^-$.

### 3.2. Geochemical Response of Reservoir Minerals

The analysis of mineral composition showed that the content of dolomite in the debris samples increased first and then decreased with the progress of the reaction, indicating that dolomite precipitated at the beginning and then gradually dissolved (Figure 6a). Quartz and illite showed the opposite trend (Figure 6a,b), decreasing first and then increasing. The plagioclase was completely dissolved after 5 days of the experiment (Figure 6b). The contents of calcite and potassium feldspar continued to increase (Figure 6b), indicating that they were newly formed minerals and continuously precipitated out of the solution. The analysis of the principal elements (Figure 6c,d) showed that the content of $SiO_2$, $Al_2O_3$ and $TFe_2O_3$ in the debris samples decreased after the reaction, while the content of CaO, MgO and the LOI increased. The content of $K_2O$ and $Na_2O$ did not change significantly.

### 3.3. Main Geochemical Processes and Their Effects on Reservoirs

Based on the changes in the water chemistry, isotopes, mineral components and major elements in the interaction experiments between the untreated lake water and the dolomite of the Wumishan formation, it is concluded that the plagioclase dissolved and released $K^+$

$Ca^{2+}$, aluminum components, $HCO_3^-$, $SiO_2$ and $H_4SiO_4$ in the initial stage of the reaction; the hydrolysis of illite released $K^+$, $HCO_3^-$ and amorphous. At the same time, the calcite, dolomite and potassium feldspar in the solution gradually became supersaturated, and then precipitated, resulting in $Ca^{2+}$ and $Mg^{2+}$ concentrations decreasing, while $SiO_2$ increased. With the progress of the experiment, illite was gradually supersaturated and precipitated, and the incomplete dissolution of dolomite and calcite led to the dissolution of dolomite in the later stage, while calcite and potassium feldspar continued to precipitate. In summary, the dolomite reservoir would be dissolved and calcite, quartz and illite minerals would be precipitated during the recharge process of lake water. For geothermal recharge, however, it is usually clay minerals that are associated with reservoir blockage or failure [51]. The literature abundantly documents that clays like illite could induce a bad effect, mainly in clastic and argillaceous sandstone reservoirs, by plugging the matrix porosity during the fine-particle migration associated with reinjection processes.

This series of experiments revealed that the dolomite reservoir of the Wumishan formation in the study area harbored only a trace amount of illite. The interaction between water and rocks facilitated the formation of illite subsequent to the recharge of lake water into the geothermal reservoir. Notably, the minimal precipitation of illite, accounting for merely 0.09% of the debris involved in the reaction, had a virtually negligible impact on the reservoir structure in the context of actual karstic geothermal exploitation. This implies that such exploitations typically exhibit high flow rates and potentially large hydraulic drains. Given the characteristics of the karstic system, the occurrence of these minimal illite precipitates would be insufficient to plug the hydraulic drains. Therefore, the presence of illite in this context is unlikely to pose a significant obstacle to the effective extraction of geothermal energy from these reservoirs.

## 4. Conclusions

According to this experimental study on the interaction between untreated lake water and carbonate reservoirs of the Wumishan formation in the Dongli Lake area, Tianjin, the feasibility and the effect on the reservoir of direct recharge with exogenous water were discussed. The results showed that:

(1) The lake water in the Dongli Lake area is Alkaline (pH = 8.3) and of the Cl·HCO3-Na type. The TDS of the water is 1618 mg/L, which is similar to the geothermal reservoir fluid, and the carbonate minerals (calcite, dolomite and aragonite) are supersaturated.

(2) In the initial stage of the untreated lake water recharge experiment, plagioclase dissolved and released alkali metals such as K, Ca, Na and $SiO_2$, resulting in the supersaturation of dolomite, calcite and potassium feldspar in the solution and precipitation. As the reaction progressed, incomplete dissolution and precipitation caused some dolomite to dissolve, while calcite continued to precipitate, eventually leading to a significant decrease in TDS. The calculated precipitation amount was 142.6 $mg·L^{-1}$, which is equivalent to a 0.09% increase in mineral mass, indicating that the impact of lake water recharge on the reservoir structure is limited.

(3) This series of experiments show that after the untreated lake water is directly recharged into the dolomite geothermal reservoir, the water–rock interaction will promote the continuous precipitation of calcite, potassium feldspar and illite. Among them, the clay mineral illite is the most important mineral that affects the porosity and permeability of the reservoir. The influence of different forms of illite on the reservoir structure is also different, but the morphological detection of illite is still underway, so its influence on the reservoir structure remains to be further determined.

(4) The simulation results of dolomite water–rock equilibrium under different environmental change conditions (varying temperatures, partial pressures of $CO_2$, pH values and initial aqueous geochemical conditions) indicate the following: At higher temperatures, the solubility of carbonates decreases, leading to easier precipitation. As the partial pressure of $CO_2$ increases, the solubility of carbonate rocks increases, making dolomite more susceptible to dissolution. A decrease in pH value results in greater

dolomite dissolution. Higher concentrations of $Ca^{2+}$ and $Mg^{2+}$ in the initial water promote dolomite precipitation, with $Mg^{2+}$ having a more significant impact.

**Author Contributions:** D.L. conducted the investigation and data collection, developed a suitable methodology and was involved in the writing of the manuscript. Y.C. and Z.F. helped in the laboratory experiment, methodology selection and data collection. Q.Z. carried out project administration, conclusion selection, and was involved in the writing, methodology selection and editing of the manuscript. L.H. and S.L. helped in data collection and participated in editing the earlier versions of the manuscript. All authors have read and agreed to the published version of the manuscript.

**Funding:** This research was funded by the Qinghai Province science and technology plan project Research on the Integration Technology of Efficient Heat Extraction and Power Generation in Dry Hot Rocks of the Gonghe Basin, Qinghai Province (2024-QY-205); the China Geological Survey project Pilot test demonstration of exploration and trial production of dry hot rock (DD20230753); and the Tianjin Science and Technology Plan Project Research on the key technology of deep geothermal reservoir recharged by rain and flood in flood season of Dongli Lake in Tianjin (23YFZCSN00390).

**Institutional Review Board Statement:** Not applicable.

**Informed Consent Statement:** Not applicable.

**Data Availability Statement:** Data is contained within the article.

**Conflicts of Interest:** The authors declare no conflicts of interest.

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
