# Peer review of "Feasibility Study on Geothermal Dolomite Reservoir Reinjection with Surface Water in Tianjin, China"

_water, doi:10.3390/w16213144_

Round 1
Reviewer 1 Report
Comments and Suggestions for Authors
Dear authors,
you analysed the chemical reactions in a dolomite geothermal reservoir, due to the reinjection of exhausted geothermal water, to improve the reservoir stability.
The paper is interesting and investigates a not-so-common topic in the geothermal literature. I appreciated it.
Please find attached my review

Author Response
Reviewer 1:
- It is written the reservoir is simulated at 90°C, however no information over the injected water temperature is given. If authors think this variable does not has significant impacts over the chemical reactions, then they should justify it.
Response: According to Ruan*, the large amount of low-temperature fluid reinjected into the geothermal reservoir has not significantly affected the temperature and chemical fields of the reservoir. Therefore, the reinjection of surface water will not adversely impact the development and utilization of geothermal resources in the area.( Ruan C X, Shen J, Li L L, Liu R G, Mou S X. Researches on the reinjection of Dongli Lake bedrock reservoir in Binhai New Area, Tianjin. Geological Bulletin of China, 2017, 36(8):1439-1449)
- Some graphs showing the evolution of chemical reactions at 5, 10 and 20 days can be useful to the reader. A forecast of the trend of chemical reactions (and maybe its stabilization) over higher periods would be appreciated.
Response: Due to the limitations in funding and equipment, this experiment has a certain exploratory nature, and the research results obtained are preliminary. The next step will involve conducting comparative experiments with a longer time series.
- The conclusions section is really short and only partially answer the main question of the work. At the end, it is hypothesized that the chemical reactions could change permeability and porosity of the reservoir, then affecting the whole geothermal work. But it is not quantified how these changes can affect the energy extraction. More emphasis and more insights over this would be very much useful to the geothermal community. I would suggest, for example, to include a simulation of the work of the geothermal reservoir, before and after the closure of pores, due to chemical reactions. Afterwards, a sensitivity analysis can be conducted, and some comments can be provided.
Response: Added. In the conclusion section, a sensitivity analysis was conducted to simulate the precipitation of carbonate rocks under different environmental change conditions. (Line 314 to 321)
- OTHER SUGGESTIONS IN THE FILE
I would ask to explain the chemistry of TFe2O3 instead of the usual Fe2O3.
Response: Traditional Fe2O3 refers solely to pure iron (III) oxide, a compound formed by the combination of iron and oxygen elements. In contrast, TFe2O3 represents a broader concept, encompassing all iron elements that may exist in the form of iron oxides, regardless of their specific chemical forms. In this test, besides the traditional Fe2O3, other complexes were also included, hence the use of TFe2O3.
I would suggest to explain the acronym LOI
Response: Added. “LOI” is an abbreviation for Loss On Ignition, with an explanation added in the text. (Line 128).
Some English typos are present throughout the text.
Response: Modified. The English has been improved.
In the Introduction, a reference about the Denmark geothermal system must be added.
Response: Added. A reference on geothermal fields in Denmark has been added (Line 43 and line 373).
In Figure 1, it would be useful to insert also a map of the location with respect to China.
Response: Added. A map of the location with respect to China has been inserted. (Line 111)

Reviewer 2 Report
Comments and Suggestions for Authors
The topic of reinjecting shallow water into a karstic reservoir for minimizing reservoir depletion is quite challenging and valuable. The authors considered as a basic idea to test experimentally the geochemical water rock interaction between the lake water and the dolomite reservoir rock versus time during reinjection. They present valuable data and results that can be improved.
However, I did some comments and proposed major revision for improving the scope of this kind of anthropogenic exploitation operation within reinjection geothermal wells.
As a general comment, there are too many literature references for a such short paper. Moreover, there is a general shift in the citation labelling in the manuscript from reference #14. Please check and correct accordingly.
Materials and Methods section
It would be valuable to have some general information about the Tianjin geothermal field like the production and reinjection temperature, the average flowrate as well as the average thermal power in MW. The reinjection pressure would be interesting or the reinjection index for example in L/s/MPa.
The terms "mining well" is not used in geothermal industry. It would be better to use the terms "production well" and consequently correct the legend of the Figure 1 (right).
Inside Figure 1 (right part), please correct "Jinzhong Rivre" in "Jinzhong River".
Sample description
First, a brief XRD apparatus description and the general analytical conditions must be mentionned like the XRD data acquisition time, type of radiation, type of diffractometer (40kV, 40 mA for example), etc....
Line105 to Line106. Whole XRD is not appropriate for identifying clay minerals like illite (dioctahedral clay). Considering the very low percentage of illite in the rock sample (0.7%), I have some doubts about their proper characterization via whole XRD alone. In general, below 5% of occurrence for a given mineral detected by XRD, it is difficult to identify clearly minerals specially for clay minerals like illite. Same remark for plagioclase with an occurrence of 0.15%. Diffractogram patterns are not shown in the paper.
Clay minerals are quite common in karstic formations like kaolinite, smectite, chlorite and illite/mica. Thus for illite identification, you need to carry out specific XRD measurements called oriented clay fraction. As you did only whole XRD, I have serious doubts about your results considering the low amount of clay in your sample (0.7%). It would have been necessary to carry out oriented clay fraction analysis to specifically determine illite minerals. Thin section description would adequaly confirmed the occurrence of illite.
From a geological point of view, it very coherent to find dolomite and quartz but more challenging to find illite and plagioclase for the reasons explained before.
Where are coming those minerals? illite is a secondary minerals or an authigenic minerals related to diagnesis or another origin? Same question about plagioclase which is more related to crystalline rocks than sedimentary carbonates.
Line 107 to Line 110. By adding all the major chemical species (Si02, MgO, CaO, Fe2O3, Al2O3, K2O and the loss of incognition, we obtained 100.33%. What is the uncertainty of the chemical analyses? Because the very low value for K2O (0.1%) is probably below the real uncertainty for this chemical method.
It is very coherent to find 30.18% of loss of incognition in such carbonate-rich rocks. CO2 content (from carbonates) represents also a large component of those looses.
Line 122 to Line 130. The English could be improved. There are too many repetitions.
Rock sample. It consists of dolomite rock debris lying in a karstic reservoir. Thus, it doesnot correspond to a reservoir rock which is generally more compact and brittle with large channels called karst. Thus by using rock debris, the impacts in terms of water rock interactions is surely modified. The rock surface is articifially enhanced by such design and probably the dissolved/precipitated phases are also enhanced. That implies that the experimental design maximize the water-rock interaction and the chemical processes. Please mention in the paper that the reaction surface being very large (due to the fact to use rock debris), the chemical results could also be enhanced.
It is not clear why you are using a stirring speed controller (rotation) in the apparatus during the experiments? In the reinjection wells, you cannot use such device. Please clarify in your text.
Line 163-Line 164. What is the thermal effect of cooling the sample at room temperature? Maybe you could preciipate some new mineralogical phases at lower temperatures than those used during the experiment?
During the experiment and specially in the stage 2 (RL-2), the pH decreases, meaning a less basic environment. In parallel, the TDS decreases as well (Figure 4c, d). This result is counter intuitive because, a less basic fluid (and thus a more acid) at stage RL-2 could easier dissolve carbonate rocks. Thus the TDS would have been increase at stage 2 and not decrease. Please comment.
Line 195-Line 199. Behaviour of carbonates with Phreeqc program. In fact, you illustrated that carbonates are always oversaturated and are not soluble when temperature increases. This is their normal geochemical behaviour when temperature raises. However, did you observed aragonite in the rock debris?
Line 200-LIne 206. You did not mention in the manuscript that the Li concentration decreases conversely to the increase of precipitation of K feldspar and illite. Generally Lithium is clearly incoporated in clays (mica, illite) and feldspar. In the same way, the abscence of barium in the fluid (Fig 4g) could be correlated to the precipitation of K-Feldspar minerals which generally trap barium.
In the discussion, it is poorly discussed how you transposed your chemical results to a real geothermal exploitation. Knowing that you are exploiting a karstic system, it means that you have high flow rate and probably large hydraulic drains (karsts). Therefore, considering the low amount of illite you could precipitate (< at 0.1%), their occurrences could not plug your hydraulic drains because of the characteristics of the karstic system.
It is well-know for litterature, that clays like illite could induce a bad effect mainly in clastic and argillaceous sandstone reservoirs by plugging the matrix porosity during fine particle migrations related to reinjection. In your case, as the reservoir is karstic, you could discussed that this effect could be minimized in a first approach.

The English could be improved a bit. There are several sections where it is hardly understandable. For example between lines 118 and 139.
I did some suggestions in the file here attached as well some comments.
Author Response
Reviewer 2:
- MATERIALS AND METHODS SECTION
It would be valuable to have some general information about the Tianjin geothermal field like the production and reinjection temperature, the average flowrate as well as the average thermal power in MW. The reinjection pressure would be interesting or the reinjection index for example in L/s/MPa.
The terms "mining well" is not used in geothermal industry. It would be better to use the terms "production well" and consequently correct the legend of the Figure 1 (right).
Inside Figure 1 (right part), please correct "Jinzhong Rivre" in "Jinzhong River".
Response: Added. Some general information about the Tianjin geothermal field have been supplemented in the MATERIALS AND METHODS SECTION. (Line 80 to 89)
Modified. Use "production well" to replace "mining well" and consequently correct the legend of the Figure 1.(Line 100 and Line 111)
Modified. Correct "Jinzhong Rivre" in "Jinzhong River".(Line 111)
- SAMPLE DESCRIPTION
First, a brief XRD apparatus description and the general analytical conditions must be mentionned like the XRD data acquisition time, type of radiation, type of diffractometer (40kV, 40 mA for example), etc....
Response: Added. A brief XRD apparatus description and the general analytical conditions have been supplemented in 2.2.1. SAMPLE DESCRIPTION. (Line 122 to 124)
Line105 to Line106. Whole XRD is not appropriate for identifying clay minerals like illite (dioctahedral clay). Considering the very low percentage of illite in the rock sample (0.7%), I have some doubts about their proper characterization via whole XRD alone. In general, below 5% of occurrence for a given mineral detected by XRD, it is difficult to identify clearly minerals specially for clay minerals like illite. Same remark for plagioclase with an occurrence of 0.15%. Diffractogram patterns are not shown in the paper.
Clay minerals are quite common in karstic formations like kaolinite, smectite, chlorite and illite/mica. Thus for illite identification, you need to carry out specific XRD measurements called oriented clay fraction. As you did only whole XRD, I have serious doubts about your results considering the low amount of clay in your sample (0.7%). It would have been necessary to carry out oriented clay fraction analysis to specifically determine illite minerals. Thin section description would adequaly confirmed the occurrence of illite.
From a geological point of view, it very coherent to find dolomite and quartz but more challenging to find illite and plagioclase for the reasons explained before. Where are coming those minerals? illite is a secondary minerals or an authigenic minerals related to diagnesis or another origin? Same question about plagioclase which is more related to crystalline rocks than sedimentary carbonates.
Response: Added. We sincerely apologize for our inadequate understanding of the limitations of the whole-rock X-ray diffraction (XRD) analysis and testing method. Due to the lack of testing facilities in our unit, we sent samples for whole-rock XRD analysis and testing to an external laboratory. The data listed in the paper are the results of those experiments. To further substantiate the presence of clay minerals in the rocks, we have included supplementary results from scanning electron microscopy (SEM) of rock thin sections in 2.2.1. SAMPLE DESCRIPTION. (Line 117 to 122)
Line 107 to Line 110. By adding all the major chemical species (SiO2, MgO, CaO, Fe2O3, Al2O3, K2O and the loss of incognition, we obtained 100.33%. What is the uncertainty of the chemical analyses? Because the very low value for K2O (0.1%) is probably below the real uncertainty for this chemical method. It is very coherent to find 30.18% of loss of incognition in such carbonate-rich rocks. CO2 content (from carbonates) represents also a large component of those looses.
Response: After communicating with the testing unit, we have learned that the reason for the summation exceeding 100% is due to the uncertainties inherent in chemical analysis. As you mentioned, the extremely low value of K2O (0.1%) is below the practical uncertainty of the chemical method used.
Line 122 to Line 130. The English could be improved. There are too many repetitions.
Response: Modified. The English has been improved. (Line 139 to 161)
Rock sample. It consists of dolomite rock debris lying in a karstic reservoir. Thus, it does not correspond to a reservoir rock which is generally more compact and brittle with large channels called karst. Thus by using rock debris, the impacts in terms of water rock interactions is surely modified. The rock surface is articifially enhanced by such design and probably the dissolved/precipitated phases are also enhanced. That implies that the experimental design maximize the water-rock interaction and the chemical processes. Please mention in the paper that the reaction surface being very large (due to the fact to use rock debris), the chemical results could also be enhanced.
It is not clear why you are using a stirring speed controller (rotation) in the apparatus during the experiments? In the reinjection wells, you cannot use such device. Please clarify in your text.
Response: Added. In order to simulate the chemical reaction process between lake water and rocks over a long period of time within a short experimental duration, we have increased the specific surface area for water-rock reaction by using rock debris. Additionally, we have employed stirring bars within the reactor to accelerate the reaction rate between water and rocks, thereby achieving a full reaction in the laboratory within a short period of time. (Line 172 to 176)
Line 163-Line 164. What is the thermal effect of cooling the sample at room temperature? Maybe you could precipitate some new mineralogical phases at lower temperatures than those used during the experiment?
Response: Limited to test conditions, sampling and analysis cannot be carried out under high temperature conditions. We believe that in the rapid cooling process, the effect of temperature changes on mineral dissolution and precipitation is minimal.
During the experiment and especially in the stage 2 (RL-2), the pH decreases, meaning a less basic environment. In parallel, the TDS decreases as well (Figure 4c, d). This result is counter intuitive because, a less basic fluid (and thus a more acid) at stage RL-2 could easier dissolve carbonate rocks. Thus the TDS would have been increase at stage 2 and not decrease. Please comment.
Response: Modified. Based on your opinion, we rechecked the experimental data and apologized to find that we had written pH 8.4 to 7.4 due to negligence, which was a counterintuitive result. We have made corrections in this article.(Line 208)
Line 195-Line 199. Behaviour of carbonates with Phreeqc program. In fact, you illustrated that carbonates are always oversaturated and are not soluble when temperature increases. This is their normal geochemical behaviour when temperature raises. However, did you observed aragonite in the rock debris?
Response: The Phreeqc simulation results list aragonite as a possible mineral. However, due to budget constraints during the actual work process, detailed rock and mineral analysis was not conducted. It is planned to carry out detailed and precise analysis of rock minerals in the next phase of work.
Line 200-Line 206. You did not mention in the manuscript that the Li concentration decreases conversely to the increase of precipitation of K feldspar and illite. Generally Lithium is clearly incoporated in clays (mica, illite) and feldspar. In the same way, the abscence of barium in the fluid (Fig 4g) could be correlated to the precipitation of K-Feldspar minerals which generally trap barium.
Response: Added. The corresponding content has been added to the text. (Line 230 to 234)
In the discussion, it is poorly discussed how you transposed your chemical results to a real geothermal exploitation. Knowing that you are exploiting a karstic system, it means that you have high flow rate and probably large hydraulic drains (karsts). Therefore, considering the low amount of illite you could precipitate (< at 0.1%), their occurrences could not plug your hydraulic drains because of the characteristics of the karstic system. It is well-known for litterature, that clays like illite could induce a bad effect mainly in clastic and argillaceous sandstone reservoirs by plugging the matrix porosity during fine particle migrations related to reinjection. In your case, as the reservoir is karstic, you could discussed that this effect could be minimized in a first approach.
Response: Added. The corresponding content has been added in 3.3. MAIN GEOCHEMICAL PROCESSES AND THE EFFECTS ON RESERVOIRS. (Line 276 to 290)
- OTHER SUGGESTIONS IN THE FILE
Line 13. “new approach”. It is not really “new” or “is new in China”. It was done in California in the Geysers geothermal field, which is the largest exploited geothermal field in the world in 2000. See for instance Kaya et al (2011) in Renewable and Sustainable Energy Reviews, 2011, who did a review and mentioned that reinjection was done in the Geysers field in 1998.
Response: Modified. Write “a new approach in China” instead of “a new approach”. (Line 13).
Line 16. “original geothermal reservoir water”. It is better to write the “native geothermal reservoir water”.
Response: Modified. Use “native geothermal reservoir water” instead of “original geothermal reservoir water”. (Line 18 to 21).
Line 18- Line 21. There is a grammatical issue in this sentence which is too long. Please rephrase. The scientific meaning is understandable, but the sentence is too long.
Response: Modified. The English has been improved. (Line 18 to 21).
Line 21 - Line 22.Plagioclase cannot release “K”. Thus “K” and K-feldspar” must be removed from this sentence or you need to mention that other minerals bearing potassium could be dissolved (like biotite).
Response: Modified. Removed “K” and K-feldspar” from the sentence (Line 22 to 23).
Precipitation of dolomite and calcite. For the dolomite precipitation, from which minerals is coming the mg? From which minerals is coming the carbonates (dissolved) for the dolomite and the calcite?
Response: The Mg for dolomite precipitation originates from Mg2+ in water, and as precipitation proceeds, the Mg content in the water decreases. The carbonates in dolomite and calcite come from dissolved CO2.
Line 24. Write “Illite” instead of “Hillite”
Response: Modified. (Line 25)
INTRODUCTION
Line 14 and others. There is a shift in the references list. For examples, when you cited reference number 14, as a reference work for reinjection in France, it is not correct. Thus, it seems that you shifted all the references in the whole manuscript. Please check properly.
Response: Modified. We have revised reference number 14 (Line 371)
Line 55 - Line 56. You cited twice the term “regulations” in the same sentence. Please adjust. No need to cite twice this term.
Response: Modified. (Line 56)
Line 82. Write “geothermal” instead of “thermal” reservoirs Line 83. Same remark.
Line 85. Same remark.
Response: Modified. (Line 92 and Line 93) The full text was checked and corrected.
Line 91. Do you mean production wells for mining wells? Line 93. Write “106m3/a”
Response: Modified. Use "production well" to replace "mining well" (Line 100)
Modified. Write “106m3/a” to replace“106m3/a” (Line 102)
Line 88 - Line 94. Please indicate how many MW thermal are produced by this geothermal field? It is more indicative than the yearly cumulative volume. It would be more interesting for the reader to know the average production temperature as well as the average production flow rate.
Response: Added. We have added the amount of thermal generated in the Dongli Lake area each year. (Line 103 to 105)
Line105-Line106. Whole XRD is not appropriate for identifying illite minerals (dioctahedral clay) with a so low percentage (0.7%). Same remark for plagioclase.For illite identification, you need to carry out specific XRD measurements called oriented clay fraction. As you did only whole XDR, I have serious doubts about your results. Moreover, the bulk diffractogram pattern is not shown in the paper. The bulk chemical analysis confirms that Al2O3 and K2O are very low and. Al2O3 is related to plagioclase, but both are inside the composition of illite.
Response: Added. We sincerely apologize for our inadequate understanding of the limitations of the whole-rock X-ray diffraction (XRD) analysis and testing method. Due to the lack of testing facilities in our unit, we sent samples for whole-rock XRD analysis and testing to an external laboratory. The data listed in the paper are the results of those experiments. To further substantiate the presence of clay minerals in the rocks, we have included supplementary results from scanning electron microscopy (SEM) of rock thin sections in 2.2.1. SAMPLE DESCRIPTION. (Line 117 to 122)
Line 116. Write “oversaturated” instead of “supersaturated”.
Response: Modified. (Line 135)
Line 122. What means “flocs”?
Response: Modified. “flocs” means “large flocculent precipitates (aluminum floc)”. The presentation has been improved. (Line 143)
Line 118-Line 139. The English could be improved in this section. It is understandable but difficult to follow properly.
Response: Modified. In 2.2.2. PRE-TREATMENT OF WATER SAMPLES BEFORE EXPERIMENTS the English has been improved. (Line 139 to 161)
Line 140. In Figure 2, what means “NF”?
Response: Added. “NF” means. “nanofiltration”. ( Line 154)
Line 159. Add a space between “20mL” and “and”, and between “100mL” and “sample”.
Response: Added. Line 187.
Line 161. What means physically 200 mesh? Which size?
Response: The sample requirements for XRD whole-rock analysis specify that the particle size should be less than 0.075mm, which corresponds to a mesh size of below 200.
Line 166-Line 167. Add a space between “immediately” and “(Table 1)”. Line 178. Add a space between “initial” and “value”.
Response: Added. Line 195 and line 206.
Line 200. Write “trace elements” instead of “microelements”
Response: Modified. Line 227.

Round 2
Reviewer 1 Report
Comments and Suggestions for Authors
Dear authors,
thank you for answering the questions.
Please pay attention at page 3, line 127. There is a typo (Tfe2O3). Moreover, I would reccommend explicit in the paper the meaning of TFe2O3.
Author Response
- Please pay attention at page 3, line 127. There is a typo (Tfe2O3). Moreover, I would recommend explicit in the paper the meaning of TFe2O3.
Response: Modified and Added. We have corrected the typo (Tfe2O3) and added an explanation for TFe2O3 in the text. (Line 128 to 130)

Reviewer 2 Report
Comments and Suggestions for Authors
The authors answered properly to all the questions/comments and took into account about in their revised manuscript.
Comments on the Quality of English LanguageProbably the English could be improved a bit. I am not a native English but I think it could be checked.
Author Response
- Probably the English could be improved a bit.
Response: Modified. Thank you for your valuable suggestions. we have checked the whole manuscript and modified some typos. The details are marked in Highlight in the revised manuscript.
